# Regioselective functionalization of aryl azoles as powerful tool for the synthesis of pharmaceutically relevant targets

Ferdinand H. Lutter [1], Lucie Grokenberger[1], Luca Alessandro Perego [2], Diego Broggini[2], Sébastien Lemaire [3], Simon Wagschal [2✉] & Paul Knochel [1✉]

Aryl azole scaffolds are present in a wide range of pharmaceutically relevant molecules. Their ortho-selective metalation at the aryl ring is challenging, due to the competitive metalation of the more acidic heterocycle. Seeking a practical access to a key Active Pharmaceutical Ingredient (API) intermediate currently in development, we investigated the metalation of 1-aryl-1H-1,2,3-triazoles and other related heterocycles with sterically hindered metal-amide bases. We report here a room temperature and highly regioselective ortho-magnesiation of several aryl azoles using a tailored magnesium amide, TMPMgBu (TMP = 2,2,6,6-tetra-methylpiperidyl) in hydrocarbon solvents followed by an efficient Pd-catalyzed arylation. This scalable and selective reaction allows variation of the initial substitution pattern of the aryl ring, the nature of the azole moiety, as well as the nature of the electrophile. This versatile method can be applied to the synthesis of bioactive azole derivatives and complements existing metal-mediated ortho-functionalizations.

[1] Ludwig-Maximilians-Universität München, Department Chemie, Butenandtstrasse 5-13, Haus F, 81377 München, Germany. [2] Discovery Product Development and Supply, Janssen Pharmaceutica, Hochstrasse 201, 8200 Schaffhausen, Switzerland. [3] Discovery Product Development and Supply, Janssen Pharmaceutica, Turnhoutseweg 30, B-2340 Beerse, Belgium. ✉email: swagscha@its.jnj.com; paul.knochel@cup.uni-muenchen.de

**N**-aryl azole scaffolds are present in several marketed and experimental drugs, such as celecoxib[1], apixaban[2], zibotentan[3], and nesapidil[4] (Fig. 1a). As part of an ongoing development program, we sought a straightforward access to N-aryl-1,2,3-triazole 1a[5]. An attractive and efficient approach to access such heterocyclic motif is the C–H functionalization of 1-aryl-1H-1,2,3-triazoles such as 2a (Fig. 1b). A well-established strategy involves transition metal-catalyzed C–H arylations[6–21]. These reactions usually require harsh conditions and often lead to bis-arylated products, which limits their practicality[6,8,12–17,22,23]. The direct deprotonation with a suitable base may be an alternative for the selective functionalization of aryl azoles. However, the regioselective metalation of the aryl ring linked to a heterocycle is challenging, due to the competitive and often favored metalation of the N-heterocycle itself[24].

A potential approach to achieve a regioselective metalation at the aryl ring is the avoidance of coordinating solvents such as THF, which competes with the nitrogen atom of the azole ring in complexation of the base[25]. Sterically hindered metal-amide bases, especially magnesium- and zinc-derived TMP-bases (TMP = 2,2,6,6-tetramethylpiperidyl) have proved to be powerful reagents for the functionalization of various (hetero)arenes[26–33]. The use of hindered metal amides in hydrocarbon solvents should thus be beneficial. In line with this concept, Hagadorn showed that TMP$_2$Zn is an excellent base for the α-zincation of various carbonyl compounds and the metalation of pyridine-N-oxide in toluene[34,35]. Similarly, Mulvey and co-workers[36–43] reported several mixed bimetallic amide bases for metalation reactions in non-coordinating hydrocarbon solvents. Herein we report a highly selective and broadly applicable magnesiation of various aryl azoles using the amide base TMPMgBu in a toluene/hexane solvent mixture and subsequent cross-couplings and electrophilic quench reactions.

## Results

**Reaction optimization.** In preliminary experiments, the reaction of 1-aryl-1H-1,2,3-triazole 2a with various metal-amide bases was examined to assess the selectivity between products **A** and **B**. The use of strong bases like TMPLi or LDA exclusively afforded the undesired metalation at the most acidic 5-position of the triazole together with large amounts of decomposition products (Fig. 2a, entries 1–2). Similarly, mixtures of **A** and **B** were obtained with TMPMgCl·LiCl or TMP$_2$Mg in THF[44–46] (entries 3–4). We turned our attention to TMPMgBu[47,48], which was conveniently prepared by treating TMP-H with commercially available Bu$_2$Mg in hexane (25 °C, 48 h), affording a clear 0.74–0.81 M solution in 94–98% yield (Fig. 2b). Unfortunately, performing the metalation of 2a in THF using TMPMgBu did not yield better results in terms of selectivity between the two metalation sites (Fig. 2a, entry 5). As mentioned above, we anticipated that the use of the highly coordinating solvent THF could hamper a selective coordination at the N(2)-atom of the triazole. We therefore switched to metal bases in hydrocarbons. While TMP$_2$Mg in toluene proved to be too reactive, leading to extensive decomposition of the starting material 2a (entry 6), TMPMgBu in toluene turned out to be highly selective, affording the desired metalated triazole **A** in 81% yield within 1 h (**A:B** = 96:4, entry 7). However, TMP$_2$Zn[34,35] or iPrMgCl.LiCl were not suitable reagents for the deprotonation of the aryl moiety of 2a (entries 8–9).

**Substrate scope.** We then examined the reactivity of the arylmetal species generated via deprotonation with TMPMgBu in the palladium-catalyzed Negishi cross-coupling (Fig. 2c). After transmetalation with ZnCl$_2$, the resulting arylzinc reagent was coupled with 4-chloro-6-methoxypyrimidine using 1 mol% of [PdCl$_2$(dppf)] (dppf = 1,1′-bis(diphenylphosphino)ferrocene) and the desired active pharmaceutical ingredient (API) intermediate 1a could be isolated in 86% yield. With these results in hand, we examined the scope of the metalation reaction using various substituted aryl triazoles (Fig. 3).

The metalation of the electron-deficient triazole 2b proceeded smoothly within 1 h at room temperature leading exclusively to the organomagnesium reagent 3b in 86% yield. The unsubstituted phenyl derivative 2c was metalated in 4 h affording 72% of the desired metal reagent 3c along with 6% deprotonation at the triazole 5-position. The electron-rich triazoles 2d–f required a prolonged metalation time of 4–6 h and furnished 3d–f in 68–77% yield. The metalation of the ortho-fluoro triazole 2g

**Fig. 1 Background and objective. a** Examples of bioactive aryl azole derivatives. **b** Retrosynthetic strategy for API intermediate 1a.

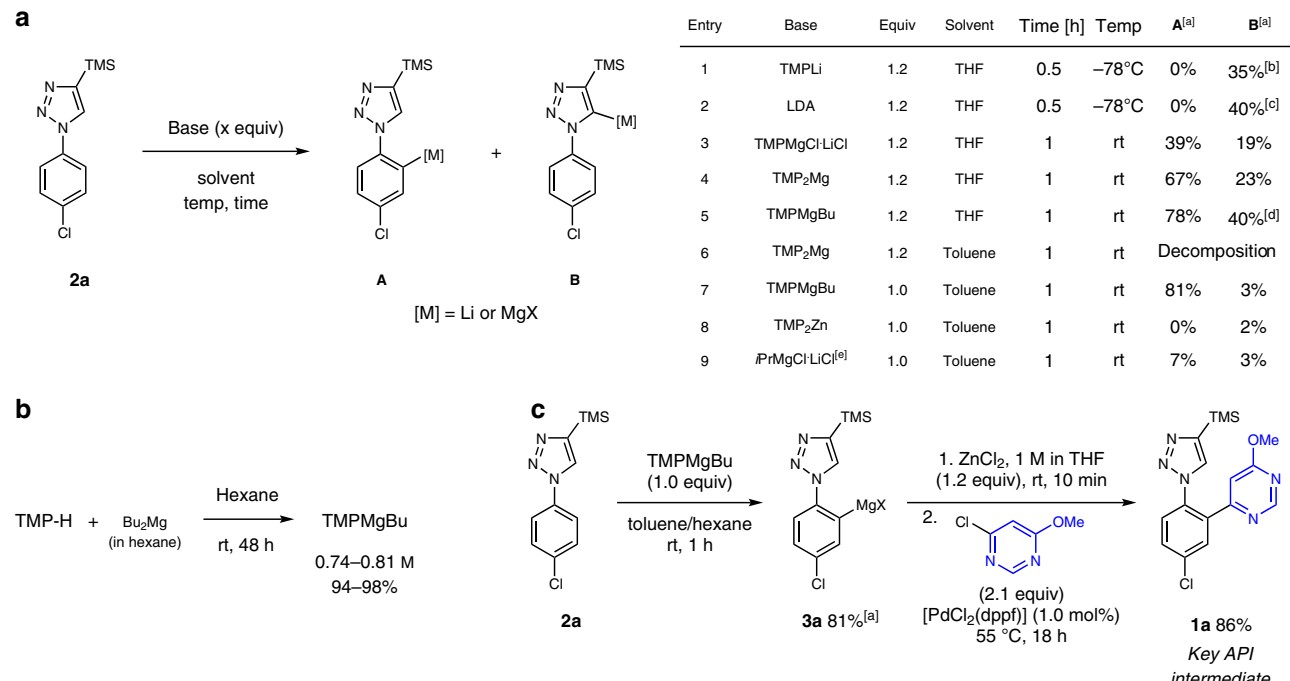

**Fig. 2 Reaction optimization. a** Screening of the metalation of **2a**. [a]Metalation yields were determined by $^1$H-NMR analysis of $D_2O$-quenched reaction aliquots. [b]Sixty-five percent decomposition. [c]Forty-three percent decomposition. [d]Including bis-metalated species. [e]In THF. **b** Preparation of TMPMgBu. **c** Pd-catalyzed cross-coupling reaction towards **1a**.

afforded **3g** in 80% yield. We did not observe any metalation ortho to either the methoxy or the fluoro moieties, indicating that the triazole unit is a much stronger directing group than those substituents. When testing other substituents at the 4-position of triazole, we found that the TMS group was key to reach high selectivity as the corresponding 4-butyl and 4-phenyl analogs afforded only mixtures of arylmagnesium species (see Supplementary Fig. 13).

Magnesium organometallics **3a–g** were then transmetalated with $ZnCl_2$ prior to their use in the cross-coupling with a variety of functionalized (hetero)aryl bromides. Palladium-catalyzed coupling reactions proceeded smoothly with several electron-rich and -deficient aryl bromides, furnishing the corresponding products **1b–f** in 75–87% yield. Remarkably, the reaction of the sterically demanding 2-bromonaphthalene led to **1g** in 74% yield. Various fluorinated aryl bromides containing a trifluoromethoxy, pentafluorosulfinyl, trifluoromethyl, or fluoro substituent were successfully applied in these couplings affording the desired arylated products **1h–k** in 70–95% yield. Furthermore, a range of heteroaryl bromides, such as pyridyl-, pyrimidyl-, indolyl-, and various thienyl- and furyl bromides were used as coupling partners leading to the corresponding products **1l–s** in 62–96% yield. Next, the metalation was extended to other aryl azoles (Fig. 4a). Treating 1-phenyl-3,5-dimethyl-1$H$-pyrazole **4a** with TMPMgBu (1.0 equiv) for 1 h afforded **5a** in 82% yield and perfect regioselectivity. Unsubstituted pyrazole **4b** was selectively metalated at the aryl moiety leading to the magnesium reagent **5b** (78% yield). Remarkably, no competitive metalation of the azole ring was observed in any case. Furthermore, 2,5-diphenyl-1,3,4-oxadiazole **4c** underwent a selective mono-magnesiation, affording **5c** in 76% yield after 2 h metalation time. The magnesiation of phenyl oxazoline **4d** proceeded within 1 h leading to the metalated product **5d** in 77% yield.

Negishi cross-couplings starting from **5a** afforded the compounds **6a–b** in 68–95% yield under the standard conditions.

Substrates containing such a 3,5-dimethylpyrazole group are of special interest, since an oxidative cleavage via ozonolysis affords the corresponding N-acetylated anilines[17]. The unsubstituted N-aryl pyrazolylmagnesium reagent **5b** was coupled with functionalized aryl bromides bearing a tosylate and nitrile group leading to the products **6c–d** in 89% and 88% yield, respectively. The reaction of **5c** with bromobenzene afforded the corresponding 1,3,4-oxadiazole **6e** in 80% yield, which is a valuable precursor for the synthesis of electroluminescent compounds[49].

Additionally, a more electron-deficient derivative was synthesized following the optimized procedure leading to **6f** in 75% yield. Finally, the cross-coupling of **5d** furnished the corresponding products **6g–h** in 91–96% yield.

The versatility of the method was shown by performing various trapping reactions of the arylmagnesium reagent **3a** with several commonly used electrophiles (Fig. 4b). Thus, a reaction with $I_2$ afforded **7a** in 98% yield and the addition of benzaldehyde or $MeSSO_2Me$ to **3a** led to the corresponding alcohol **7b** or thioether **7c** in 86% and 75% yield, respectively. A transmetalation with $CuCN·2LiCl$ and subsequent reaction with benzoyl chloride or an allyl bromide derivative afforded **7d–e** in 62–77% yield.

**Late-stage diversification.** Various late-stage modifications were performed to demonstrate the synthetic utility of the cross-coupling products (Fig. 5). The TMS group could be easily removed using TBAF giving access to unsubstituted triazole **8** in 91% yield. Treating **1h** with TMPMgBu for 2 h in toluene led to the arylmagnesium reagent **9** in 80% yield. After transmetalation with $ZnCl_2$, a palladium-catalyzed cross-coupling with 5-bromo-N-methyl indole afforded the bis-arylated triazole **10** in 88% yield. The reaction of **1h** with 1,3-dibromo-5,5-dimethylhydantoin furnished the corresponding bromide **11** in 93% yield. A palladium-catalyzed Suzuki-cross-coupling of **11** with an aryl-boronic acid allows the smooth functionalization of the triazole moiety, affording **12** in 86% yield.

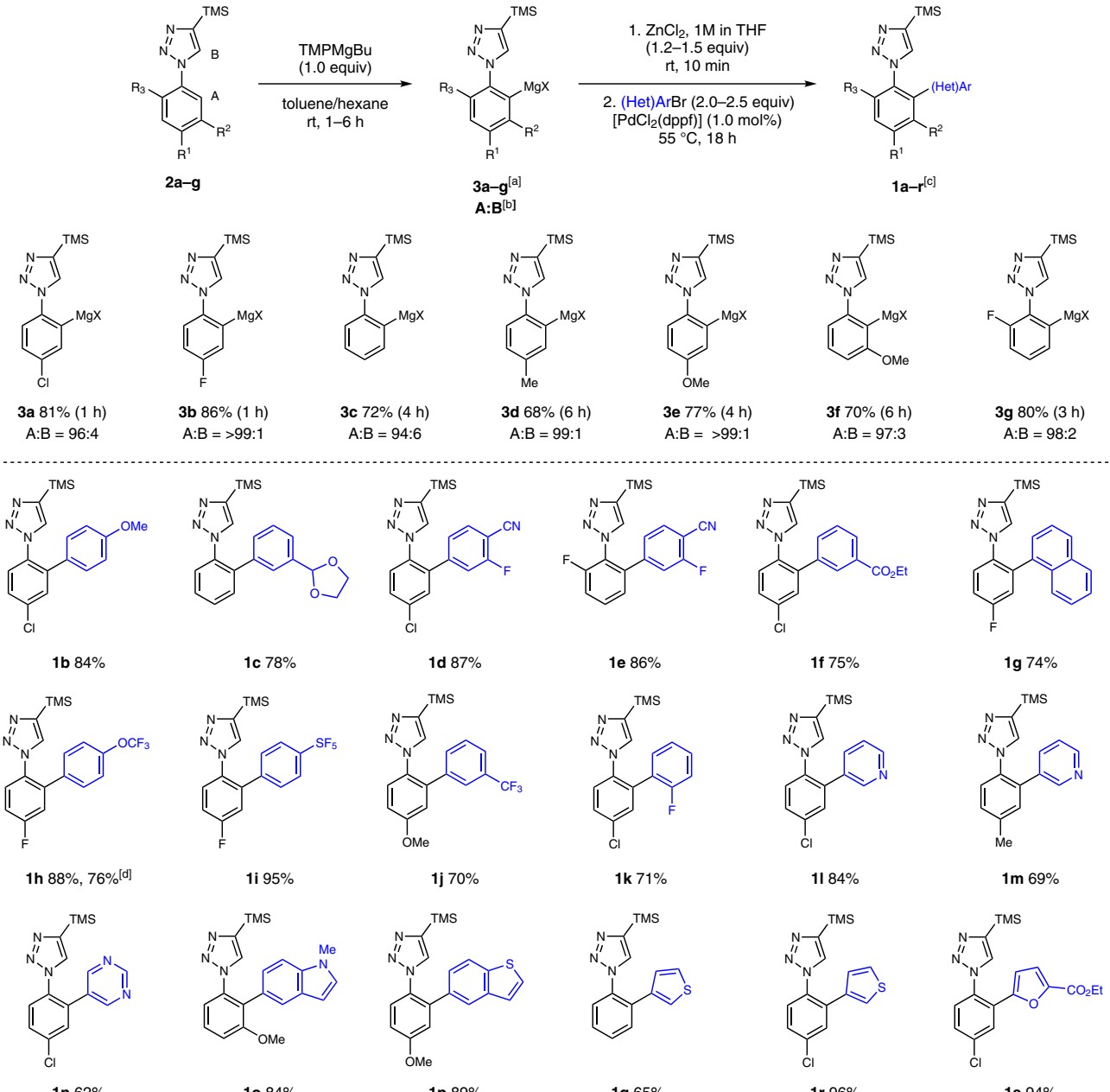

**Fig. 3 Metalation of various aryl triazole derivatives and scope of the subsequent palladium-catalyzed cross-coupling.** Experiments were performed on a 0.5 mmol scale. [a]Metalation yields were determined by $^1$H-NMR analysis of D$_2$O-quenched reaction aliquots. Metalation time in brackets. [b]Metalation ratio in [%] between regioisomers of type A and B. [c]All yields refer to isolated compounds. [d]Reaction performed on a 5 mmol scale.

**Mechanistic probes**. We then sought to gain a deeper understanding of the metalation and cross-coupling steps. It is known that commercially available Bu$_2$Mg solutions are mixtures of $n$-butyl and $s$-butyl magnesium species. Analysis of an iodolyzed sample revealed a 60:40 ratio of $n$-butyl and $s$-butyl moieties present in Bu$_2$Mg, and the same ratio was found in TMPMgBu. Interestingly, Bu$_2$Mg in toluene/hexane was also an excellent base to selectively deprotonate **2a** affording ortho-magnesiation in 93% yield (Fig. 6). The resulting mixture mainly contained ArMg ($n$-Bu) and ArMg($s$-Bu) (89% and 4%, respectively). However, after transmetalation with zinc chloride, only 28% of the desired cross-coupling product **1b** were obtained together with 88% of 4-butyl-anisole (**13a**), resulting from the cross-coupling of the $n$-butyl residue. This observation accounts for the superiority of

TMPMgBu to Bu$_2$Mg in the metalation/cross-coupling sequence: the use of TMPMgBu limits the formation of the ArMgBu and thus after transmetalation ArZnBu, which preferentially transfers the butyl group to Ar'Br, forming the byproduct Ar'Bu **13a**.

## Discussion

In conclusion, we have described a highly regioselective magnesiation of various aryl azoles using a hindered mixed magnesium amide base, TMPMgBu, in toluene/hexane at room temperature. Subsequent palladium-catalyzed cross-couplings with a variety of (hetero)aryl bromides or trapping with electrophiles afforded polyfunctionalized aryl azoles in good to excellent yields. This methodology could be applied to the synthesis of a key API

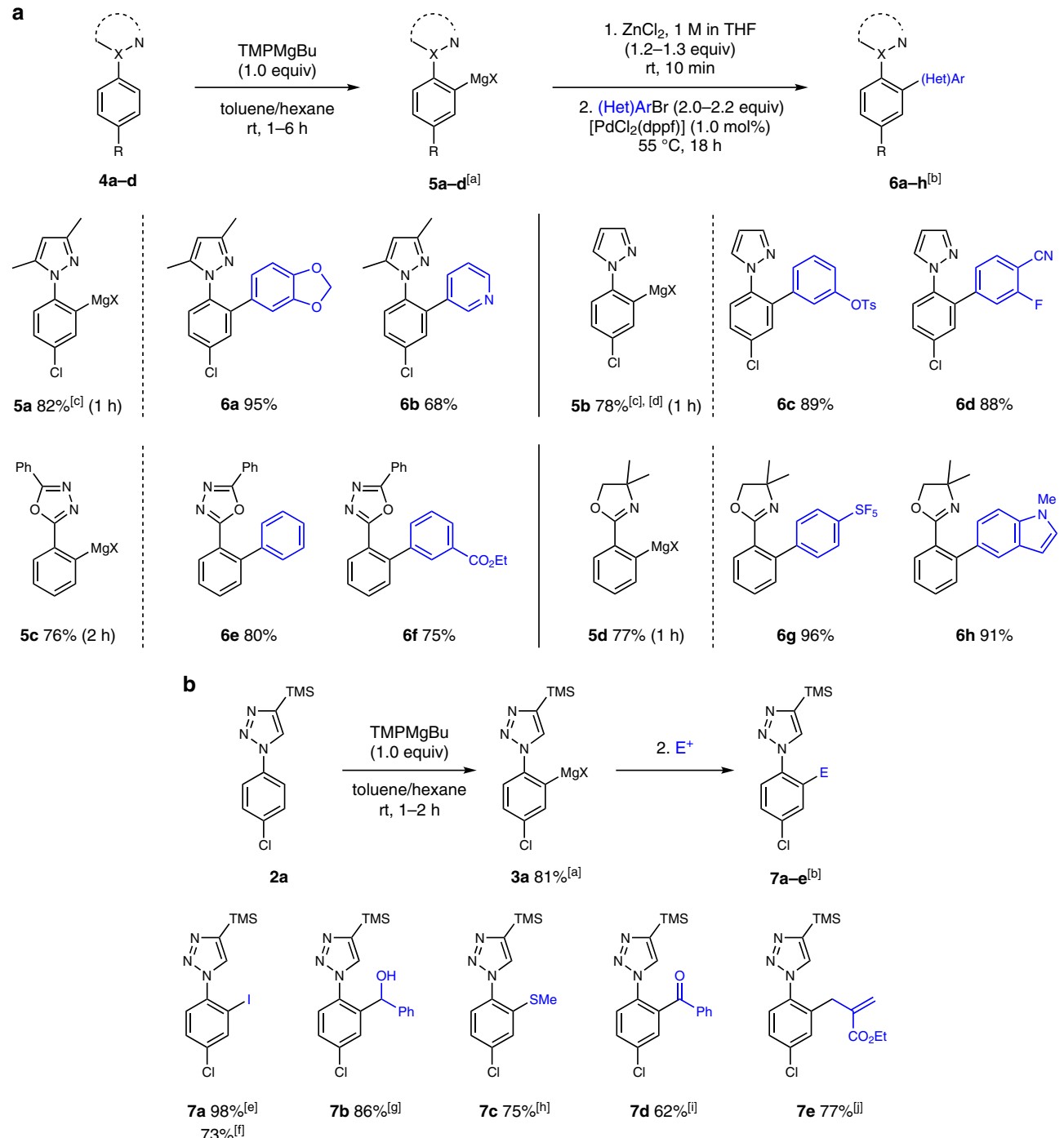

**Fig. 4 Substrate scope of aryl azoles and electrophiles. a** Metalation of various aryl triazole derivatives and scope of the subsequent palladium-catalyzed cross-coupling. [a]Metalation yields were determined by [1]H-NMR analysis of D₂O-quenched reaction aliquots. Metalation time in brackets. [b]All yields refer to isolated compounds. [c]No metalation at the heterocycle was observed. [d]Performing the reaction with TMPLi exclusively led to metalation of the azole moiety. **b** Trapping of magnesium reagent **3a** with various electrophiles. [e]I₂ (4.3 equiv). [f]Metalation of **2a** with Bu₂Mg (vide infra), then I₂ (4.3 equiv). [g]Benzaldehyde (2.5 equiv). [h]MeSSO₂Me (2.5 equiv). [i]Transmetalation with CuCN•2LiCl (3.0 equiv), then benzoyl chloride (2.5 equiv). [j] Transmetalation with CuCN•2LiCl (3.0 equiv), then ethyl 2-(bromomethyl)acrylate (2.5 equiv).

intermediate and several late-stage modifications demonstrated the versatility of the resulting products. Mechanistic experiments highlighted the key role of TMP for the reactivity of the resulting organomagnesium reagents in cross-coupling reactions.

## Methods

**Preparation of arylmagnesium reagent 3a.** Aryl triazole **2a** (126 mg, 0.5 mmol, 1.0 equiv.) was placed in a dry and argon-flushed 10 ml Schlenk tube equipped with

a magnetic stirring bar and a septum and was suspended in toluene (0.5 ml, 1.00 M). TMPMgBu (0.67 ml, 0.75 M, 1.0 equiv) was added and the mixture was stirred for 1 h affording the magnesium reagent **3a** in 81% yield.

**Palladium-catalyzed cross-coupling.** **3a** was transmetalated with a ZnCl₂ solution (0.5 ml, 1.00 M in THF) and THF (1.0 ml) was added. A dry and argon-flushed Schlenk tube, equipped with a magnetic stirring bar and a septum, was charged with Pd(dppf)Cl₂ (1.0 mol%, 0.005 mmol, 3.7 mg) and 1-bromo-4-methoxybenzene (0.850 mmol, 159 mg, 2.10 equiv) was added. The freshly

**Fig. 5 Late-stage modifications.** [a]Metalation yields was determined by $^1$H-NMR analysis of $D_2O$-quenched reaction aliquots. All other yields refer to isolated compounds. HetAr = N-Me-5-indolyl, Ar$^1$ = 4-OCF$_3$-C$_6$H$_4$, Ar$^2$ = 4-Cl-C$_6$H$_4$.

**Fig. 6 Mechanistic probes.** [a] Metalation yields were determined by $^1$H-NMR analysis of $D_2O$-quenched reaction aliquots. [b] Yields were determined by GC analysis of iodolyzed aliquots using undecane as an internal standard. Ar = 5-Cl-2-(4-TMS-1H-1,2,3-triazol-1-yl)-C$_6$H$_3$. Ar′ = 4-MeO-C$_6$H$_4$.

prepared arylzinc reagent was added and the reaction mixture was placed in an oil bath at 55 °C. After 16 h, saturated aq. NH$_4$Cl solution (5 ml) was added, the phases were separated, and the aqueous phase was extracted with EtOAc (3 × 25 ml). The combined organic layers were dried over MgSO$_4$. The solvents were removed under reduced pressure and the crude product was subjected to column chromatography.

## Data availability
The authors declare that the data supporting the findings of this study are available within the paper and its Supplementary Information files.

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

## Acknowledgements

We thank Janssen Pharmaceutica and Bristol-Myers Squibb for the funding of the project. We thank the Deutsche Forschungsgemeinschaft (DFG) and the Ludwig-Maximilians-Universität München for financial support. We also thank Albemarle Lithium GmbH (Hoechst, Frankfurt) for the generous gift of chemicals.

## Author contributions

F.H.L., L.G., and L.A.P. performed and analyzed the experiments F.H.L., D.B., S.L., S.W., and P.K. designed the experiments. F.H.L., L.A.P., S.W., and P.K. prepared the manuscript with contributions of all authors.

## Funding

## Competing interests

The authors declare no competing interests.
