## [Peer Review File · Nature Communications]

REVIEWER COMMENTS

Reviewer #1 (Remarks to the Author):

This manuscript highlights the development and utilization of a novel organometallic base, TMPMgBu, in the ortho-magnesiation of aryl azoles. Metalation is followed by various functionalization approaches, including transition metal-catalyzed cross-coupling and addition to electrophiles. This novel base is a valuable addition to the synthetic toolbox and will likely find application in the selective metalation of additional scaffolds in the future. In addition, the preparation appears to be scalable and straightforward.

A minor point of concern is the need for subsequent transmetalation to ZnCl₂ to enable Pd-catalyzed Negishi coupling. Did the authors attempt ortho metalation with a TMPZn base, in a hydrocarbon solvent system, at room temperature? This might eliminate the need for a transmetalation step in the arylation approach. The Supporting Information document provides one example with a TMPZn base at 0 °C, but higher temperatures may be required to drive zincation. I recommend one additional comparison utilizing TMP₂Zn.2MgCl₂.2LiCl in toluene at room temperature to further highlight the unique selectivity of the novel base.

Kind Regards,

Melodie Christensen

Reviewer #2 (Remarks to the Author):

Knochel and Co-workers report on an ortho-selective metalation of aryl azoles using TMPMgBu in a non-coordinating solvent, followed by a subsequent Negishi cross coupling. This approach to arylate C-H bond uses mild conditions and affords pharmaceutically relevant aryl azole in excellent yield.

When aryl triazoles are used, the methodology displays a highly selective metalation at the ortho position site despite the presence of the more acidic proton on the 5-position of the triazole moiety. The authors demonstrate that TMPMgBu is the best metal-amide base to achieve a high level of regioselectivity with toluene as solvent. However, a TMS substitution at the 4- position of the azole is required to reach a high level of regioselectivity (the authors demonstrate that the TMS-substituent could be easily removed). This metalation was also applied to other aryl azoles. Subsequently transmetalation followed by Negishi coupling was performed to delivered aryl azole derivatives.

This work is complementary to the ruthenium catalyzed direct C-H arylation protocol by Ackermann (Org. Lett. 2018, 10(11), 2299) using similar starting material and delivering similar products. Compared to Ackermann's work, this protocol is much milder as highlighted by the authors, however it is practically/experimentally more complex.

I believe that this protocol is useful for the community, however it is not disruptive enough in terms of product scope compared to Ackermann's report to be published in Nature Communication.

Few points or questions to address to the authors:

-Does the metalation occur under non-reversible conditions?

-Papers to be cited:

Ackermann (Org. Lett. 2018, 10(11), 2299)

Ballesteros (Tett. 2009, 65(22), 4410)

- In table 1, it would be interesting to see the regioselectivity outcome with LDA and $i\text{PrMgCl}\cdot\text{LiCl}$.
- It would be interesting to submit N-Aryl pyrazolymagnesium (4b) to TMPLi in order to compare with 2a (entry 1, table 1.) to see the effect of the pyrazol on the regioselectivity.
- It would be interesting to comment on the preferred directing group ability of the triazole compare to OMe and F.

Reviewer #3 (Remarks to the Author):

New conditions involving the challenging regioselective functionalization of the aryl ring of arylazoles has been developed in this manuscript. The strategy relies on the use of a new tailored magnesium amide TMPMgBu in toluene/hexane solvents followed by a Pd-catalyzed Negishi cross-coupling.

This paper represents an important breakthrough for the synthesis of polyfunctionalized arylazoles in good to excellent yields and has a number of merits:

An easy access to pharmaceutically relevant molecules.

The large scope of the method with functionalization of aryl-triazoles, -pyrazole, -oxadiazole, -oxazoline with a variety of (hetero)aryl bromide and commonly used electrophiles.

An explanation for the use of TMPMgBu through mechanistic experiments.

Overall, this contribution is suitable for publication in Nature Communications due to the reasons listed above, after considering the following minor revisions :

- 1) In many references all the authors are not mentioned and « et al » is used
- 2) P8 : as the iodolized sample of Bu_2Mg gave a 60:40 ratio of n-butyl and s-butyl, how do you explain that the magnesiation of 2a with Bu_2Mg gave a different ratio of $\text{ArMg}(n\text{Bu})$ and $\text{ArMg}(s\text{Bu})$ of 89 :4.
- 3) P8 : Interestingly, Bu_2Mg in toluene/hexane (instead of only hexane) was also an excellent base to selectively deprotonate 2a (instead of 1a)
- 4) P5 :to other aryl azoles (Scheme 2a instead of 3a)
- 5) P5 :..... leading to the magnesium reagent 5b (78% instead of 76%)

Reviewer 1

1. Did the authors attempt ortho metalation with a TMPZn base, in a hydrocarbon solvent system, at room temperature? This might eliminate the need for a transmetalation step in the arylation approach. The Supporting Information document provides one example with a TMPZn base at 0 °C, but higher temperatures may be required to drive zincation. I recommend one additional comparison utilizing TMP₂Zn.2MgCl₂.2LiCl in toluene at room temperature to further highlight the unique selectivity of the novel base.

TMP₂Zn.2MgCl₂.2LiCl is not soluble in toluene. Thus, TMP₂Zn without further salts was used for the experiment. The reaction of TMP₂Zn in toluene with 2a did not afford any deprotonation product at room temperature. This result was added to table 1 and the following sentence was added to the manuscript “However, TMP₂Zn^{34,35} or iPrMgCl.LiCl were not suitable reagents for the deprotonation of the aryl moiety of 2a (entries 8-9).”

Reviewer 2

1. Does the metalation occur under non-reversible conditions?

We propose, that the metalation occurs under non-reversible conditions. The ratio between the metalation of the aryl and theazole moiety barely changes after prolonged reaction times or at higher temperatures (see SI9 entries 14-17).

2. Papers to be cited: Ackermann (Org. Lett. 2018, 10(11), 2299); Ballesteros (Tett. 2009, 65(22), 4410)

The corresponding papers were included in the manuscript (footnotes 21 and 24)

3. In table 1, it would be interesting to see the regioselectivity outcome with LDA and iPrMgCl.LiCl.

The results were added to table 1 (entries 1 and 9). The reaction with LDA mainly led to deprotonation of the unwanted position B (70 %) together with decomposition of the starting material 2a. The reaction with iPrMgCl.LiCl, however only led to trace amounts (7%) of deprotonation product at position A. The text describing table 1 was adapted accordingly.

4. It would be interesting to submit N-Aryl pyrazolylmagnesium (4b) to TMPLi in order to compare with 2a (entry 1, table 1.) to see the effect of the pyrazol on the regioselectivity.

The reaction of 4b with TMPLi exclusively led to deprotonation of the pyrazole moiety (75 %) but no metalation at the aryl ring, underlining again the unique reactivity of TMPMgBu. A footnote was added in Scheme 2: “[b]performing the reaction with TMPLi exclusively led to metalation of theazole moiety.”

5. It would be interesting to comment on the preferred directing group ability of the triazole compare to OMe and F.

We have added the following sentences: “We did not observe any metalation ortho to either the methoxy or the fluoro moieties, indicating that the triazole unit is a much stronger directing group than those substituents.”

Reviewer 3

1. In many references all the authors are not mentioned and « et al » is used

According to the Nature References style, all authors should be included in reference lists unless there are six or more, in which case only the first author should be given, followed by 'et al.'.

2. P8 : as the iodolized sample of Bu₂Mg gave a 60:40 ratio of n-butyl and s-butyl, how do you explain that the magnesiation of 2a with Bu₂Mg gave a different ratio of ArMg(nBu) and ArMg (sBu) of 89 :4.

According to our observations, the more reactive sec-butyl magnesium reagent is performing a fast deprotonation reaction leading to almost full consumption of s-BuMgX. The rest of the deprotonation in order to achieve 93% of 2a is performed by the n-BuMgX species.

3. P8 : Interestingly, Bu₂Mg in toluene/hexane (instead of only hexane) was also an excellent base to selectively deprotonate 2a (instead of 1a); P5 :to other aryl azoles (Scheme 2a instead of 3a); P5 :..... leading to the magnesium reagent 5b (78% instead of 76%)

All suggested corrections were made in the revised manuscript.

Changes made by the authors:

Table 1: *the metalation yield of entry 1 was corrected. The metalation occurs regioselectively at the azole moiety, however 65% decomposition of 2a were observed. A footnote was added: “[b]65% decomposition.” This underlies even more the unique metalation property of TMPMgBu.*

Scheme 4: *the values for the distribution of ArMgBu and ArMgTMP were corrected.*

We apologize for these mistakes.